# Tracking the Epidemiologic Shifts in Hepatitis A Sero-Prevalence Using Age Stratification: A Cross-Sectional Study at Jordan University Hospital

**DOI:** 10.3390/pathogens10091081

**Published:** 2021-08-26

**Authors:** Nariman Kareem, Khaled Al-Salahat, Faris G. Bakri, Yaser Rayyan, Azmi Mahafzah, Malik Sallam

**Affiliations:** 1Department of Pathology, Microbiology and Forensic Medicine, School of Medicine, The University of Jordan, Amman 11942, Jordan; narimana.kareem@gmail.com (N.K.); khalid_1986_6@yahoo.com (K.A.-S.); mahafzaa@ju.edu.jo (A.M.); 2Department of Clinical Laboratories and Forensic Medicine, Jordan University Hospital, Amman 11942, Jordan; 3Department of Internal Medicine, School of Medicine, The University of Jordan, Amman 11942, Jordan; fbakri@ju.edu.jo (F.G.B.); y.rayyan@ju.edu.jo (Y.R.); 4Infectious Diseases and Vaccine Center, The University of Jordan, Amman 11942, Jordan; 5Gastroenterology and Liver Division, Department of Internal Medicine, Jordan University Hospital, Amman 11942, Jordan

**Keywords:** epidemiology, immunization, vaccine, sanitation, seroprevalence, Middle East

## Abstract

The study of hepatitis A virus (HAV) epidemiology and sero-prevalence has important public health implications. Changes in the epidemiology of hepatitis A can result in a larger pool of susceptible persons in countries with improved sanitation and hygienic conditions if vaccination is not provided. The aim of this study was to investigate the prevalence of HAV immunoglobulin G (IgG) in Jordan. In addition, we aimed to identify the potential differences in HAV sero-prevalence based on age, among other variables. We recruited the study participants at Jordan University Hospital in Amman, Jordan, during October 2020–June 2021. Assessment of participants’ socio-demographic variables was done using a paper-based questionnaire. Testing for HAV IgG was based on a competitive enzyme linked immunosorbent assay (ELISA). The study population comprised 360 individuals with a median age of 18 years. The overall sero-prevalence of HAV in our study sample was 38.3%. Divided by age, the sero-prevalence of HAV was 8.2%, 12.3%, and 20.8% among individuals aged 10 years or less, 15 years or less, and 30 years or less, respectively. The estimated age at mid-population immunity was between 21 and 30 years. Besides age, individuals residing outside the Central region of Jordan had a significantly higher HAV sero-prevalence. Additionally, the use of filtered municipal water was associated with a lower sero-prevalence of HAV compared with the use of unfiltered municipal water among individuals aged 15 years or less. The results of this study suggest an intermediate to low endemicity of HAV in Jordan. An epidemiologic shift of HAV sero-prevalence with a declining rate of positivity for HAV IgG was noticed in this study. This highlights the importance of the recently introduced HAV vaccination in Jordan. Future research to evaluate the public health benefits of HAV vaccination in Jordan is recommended.

## 1. Introduction

Hepatitis A is an acute viral infection of the liver that is mainly transmitted via the fecal−oral route [1]. Each year, this infection results in approximately 1.4 million cases of clinically apparent disease worldwide, with higher rates of inapparent infections [1,2].

The clinical manifestations of hepatitis A are largely determined by the age of the infected individuals [3]. Specifically, less than a third of infected young children are reported to develop symptomatic disease, whereas the majority of infected adults are expected to develop conspicuous (occasionally severe) acute hepatitis manifestations (fever, fatigue, anorexia, abdominal pain, jaundice, nausea, and vomiting) with markedly elevated liver enzymes [3,4,5].

The transmission of hepatitis A virus (HAV) through the fecal−oral route implies that hygienic and sanitary conditions are important factors in the epidemiology of the disease, with a higher sero-prevalence in developing countries [2,6]. On the contrary, the delayed exposure to HAV in developed countries can result in a higher possibility of large symptomatic outbreaks of hepatitis A [7,8]. Large-scale outbreaks of hepatitis A are frequently associated with the consumption of contaminated water or food (e.g., leafy green vegetables, frozen fruits, and ready-to-eat salads) [9,10,11].

In addition, the risk of hepatitis A extends to involve the occurrence of fulminant disease with acute liver failure in about 1% of the affected adult population [12]. Thus, elucidation of the epidemiologic features of hepatitis A can be helpful to assess the potential risks of such infection, as well as the need to implement adequate preventive measures (e.g., vaccination) [13].

An important determinant of HAV sero-prevalence is the socio-economic conditions of the region or country under study, which hints at the prevailing hygienic and sanitary conditions [2,14]. Hence, countries with crowding, low access to clean water, and lack of proper sewage disposal are expected to display a high HAV endemicity and sero-prevalence [15,16]. Nevertheless, these countries represent a low burden of hepatitis A, considering the natural history of the disease where a majority of adults are already immune in such countries [2]. On the other hand, it is plausible to assume that hepatitis A represents a higher burden in developed countries—with potential for outbreaks—considering the larger pool of susceptible individuals (if vaccination is not implemented) [15,17,18].

The World Health Organization (WHO) advocates two approaches to measure HAV sero-prevalence, which are based on the detection of anti-HAV immunoglobulin G (IgG) antibodies to determine the endemicity of HAV in countries with a majority of non-vaccinated individuals [19]. The first approach relies on determining the prevalence in the entire population, while the second utilizes an age-specific approach [19]. For the former method, the endemicity of HAV is classified into a high level if the prevalence exceeds 50% of the population, intermediate endemicity (prevalence of 15–50%), and low endemicity if the prevalence is less than 15% in the studied population [19,20]. The later approach classifies endemicity as follows: high (≥90% by age 10 years), intermediate (≥50% by age 15 years, with <90% by age 10 years), low (≥50% by age 30 years, with <50% by age 15 years), and very low (<50% by age 30 years) [21]. The age-specific approach is considered to be more precise and gives a more accurate estimation of the sero-prevalence [21].

Based on the aforementioned points, the concept of “the paradox of HAV epidemiology” can be easily fathomable. This concept describes the paradoxical relationship between HAV endemicity and burden of the disease [18,20]. High HAV endemicity indicates the occurrence of infection in early childhood, which is associated with a low burden of hepatitis A. On the contrary, low HAV endemicity indicates a higher fraction of susceptible adult population with subsequent risk for severe acute manifestations (fulminant hepatitis) and the potential for large outbreaks [22].

The prevention of hepatitis A relies on several measures: use of safe clean drinking water, safe disposal of sewage, and following proper personal hygiene practices (e.g., regular handwashing) [1]. However, vaccination against HAV can be viewed as the mainstay preventive measure [23]. So far, several inactivated vaccines have been licensed for the prevention of hepatitis A with excellent efficacy and safety profiles [24]. Based on the Centers for Disease Control and Prevention (CDC), HAV vaccination is recommended for children aged 12−23 months, and for children and adolescents aged 2–18 as a catch-up vaccination [24].

In Jordan, a few epidemiologic studies on the prevalence of HAV have previously been conducted [25,26]. The latest community-based study, which was conducted in 2008, indicated an intermediate endemicity of HAV in Jordan with an overall sero-prevalence of 60% [26]. Additionally, this study displayed the epidemiologic shift of HAV towards intermediate endemicity in the country [26]. A recent study in Jordan suggested that hepatitis A vaccination covering one-year old children could be a proper cost-saving interventional measure needed to address the potential public health implication of hepatitis A in the country [27].

The objectives of the current study were (1) to assess the current epidemiological patterns of hepatitis A among individuals visiting Jordan University Hospital, (2) to estimate HAV sero-prevalence among individuals of different age groups, and (3) to assess the potential variables (age, sex, socio-economic status, source of drinking water, etc.) that might be related to a higher sero-prevalence of HAV in each age group.

## 2. Materials and Methods

### 2.1. Study Design

This is a cross-sectional study that was conducted at Jordan University Hospital (JUH) during October 2020 to June 2021. Despite the relevance of random sampling, which is the appropriate method for the estimation of the sero-prevalence of an infectious agent in a certain region, we opted to use convenience sampling due to the current coronavirus disease 2019 pandemic and its associated closure of schools and universities.

To investigate the occurrence of an epidemiologic shift in HAV sero-prevalence in Jordan, we used data from a previously published study in Jordan that was conducted in 2008 [26]. Age stratification in the previous study was done as follows: 2–4 years, 5–9 years, 10–14 years, 15–19 years, and ≥20 years [26].

### 2.2. Study Participants

The recruitment of the potential study participants was done at the Outpatient Clinics Laboratory (Phlebotomy Laboratory, JUH, Amman, Jordan). The visitors were approached with a brief introduction about the study and its purposes. If the potential participant (or parent/guardian in individuals less than 18 years old) agreed to participate in this study, then written, informed, and signed consent was obtained. This was followed by filling a paper-based questionnaire (by the first author (N.K.) after asking the questions verbally) and obtaining a serum sample for the serologic assessment of HAV (using plain blood collection tubes).

### 2.3. Ethical Considerations

The current study was approved by the institutional review board (IRB) at JUH (reference number: 10/2020/8595; decision number 99/2020). Written and signed informed consent was obtained from all participants. All collected data were treated with confidentiality.

### 2.4. Paper-Based Questionnaire

The questionnaire involved items assessing participant’s age, sex, nationality, monthly income (<500 Jordanian Dinars (JOD), 500–1000 JOD, or > 1000 JOD), educational level, main source of drinking water, and number of household contacts (Appendix A).

### 2.5. Source of Drinking Water

The self-reported source of drinking water was divided into “clean” sources (mineral water, municipal water filtered at home, or water filtered by a local provider) vs. “unclean” sources (unfiltered municipal water or stream/well water).

### 2.6. Patterns of Fast Food Consumption among the Study Participants

In this study, we proposed a scale to assess the frequency of fast (junk) food consumption. This subjective scale was based on asking the participant to self-report on a scale from 1–10 the weekly frequency of fast food consumption (food purchased at restaurants including items like falafel and shawarma), with a score of 1 indicating the absence of consuming fast food, and 10 indicating daily consumption of fast food. For the rest of the manuscript, this scale will be abbreviated as the triple-F score (3F score).

### 2.7. Detection of HAV IgG Using ELISA

The qualitative determination of HAV IgG was based on an enzyme linked immunosorbent assay (ELISA) kit (Fortress Diagnostics, Antrim, UK). Briefly, this kit is based on solid phase one-step incubation competitive principle ELISA method. The quality and validity of ELISA testing was ensured by meeting the manufacturer’s guidelines regarding the optical density (OD) values for the positive and negative controls together with the OD value of the blank. The OD values of the blank, negative controls and positive controls were valid.

Results for the participants were calculated by relating each sample’s OD value to a cut-off value, which was based on the kit instructions, indicating that the OD value for the weak positive specimens was 0.450 (coefficient of variation (CV): 8.9%, resulting in OD value of the weak positive between 0.430 and 0.470), while the OD value for the negative control was 2.400 (CV: 5.5%, resulting in OD value of the negative control between 2.333 and 2.467). Thus, we calculated a factor to consider the inter-assay variability in OD results of the negative control. This cut-off value was based on the ratio between the upper ends of OD values (considering the competitive ELISA principle and the reciprocal relationship between the OD values and positivity of the sample) for the negative control and the weak positive results, which was calculated as 2.467/0.470 = 5.249.

Then, for each batch of ELISA testing, we used this factor to calculate the cut-off value specific for that run by dividing the OD value for the negative control of that run by the calculated cut-off factor (5.249). For example, the first run had a mean OD for the negative control of 1.682, which was used to calculate the cut-off value for the positive samples as follows: 1.682/5.249 = 0.320. Thus, any sample with OD ≤ 0.320 was regarded as a positive sample, while any sample with OD > 0.320 was considered as a negative sample.

### 2.8. Statistical Analysis

All statistical analyses in this study were conducted using IBM SPSS Statistics for Windows, Version 22.0. Armonk, NY: IBM Corp. Chi-squared test (χ^2^ test), Mann−Whitney *U* test (M–W), and linear-by-linear test for association (LBL) were used as appropriate. The cut-off value that was used to determine the statistical significance was *p* < 0.050. The correlation between HAV IgG positivity (positive vs. negative) and source of drinking water (filtered municipal water vs. unfiltered municipal water) was evaluated using multinomial logistic regression, with the following covariates: age, sex, region of residence, and monthly income.

Sample size determination was based on calculations done via Epitools-Epidemiological Calculators available freely online (https://epitools.ausvet.com.au/, access on 18 August 2021). The minimum required sample size was determined at 282 based on the following parameters in “Sample size to estimate true prevalence”: Assumed true prevalence = 50.0%, sensitivity and specificity = 0.99, confidence level = 0.90, and precision = 0.05.

## 3. Results

### 3.1. General Features of the Study Population

The final number of individuals who agreed to be part of this study was 360, with female predominance (*n* = 212, 58.9%). The median age for the study population was 18 years (mean = 24, interquartile range = 10–29 years, range = 3–84 years). About two-thirds of the study participants resided in Amman governorate (*n* = 234, 65.0%), followed by Zarqa (*n* = 51, 14.2%) and Balqa (*n* = 40, 11.1%). The characteristics of the study population are summarized in (Table 1).

### 3.2. Assessment of the Overall HAV Sero-Prevalence

The overall sero-prevalence of HAV in our study sample was 38.3% (95% CI: 33.5–43.5%). The sole variable that was associated with a higher HAV sero-prevalence was the older age (*p* < 0.001, M–W). Other variables yielded statistically non-significant differences (sex, nationality, educational level, monthly income, source of drinking water, and 3F score).

Divided by age, the sero-prevalence of HAV among the study participants aged 10 or less was 8.2% (95% CI: 4.2–15.4%). For participants aged 15 or less, the sero-prevalence of HAV was 12.3% (95% CI: 8.0–18.4%), while the prevalence among those aged 30 or less was 20.8% (95% CI: 16.4–26.0%).

Stratified by the following age groups, 3–8 years, 9–14 years, 15–20 years, 21–30 years, 31–40 years, 41–50 years, 51–60 years, and >60 years, the sero-prevalence of HAV showed a significant increase from 7.7% among the youngest age group (3–8 years) to 58.3% among those aged 21–30 years, reaching 100% among those aged 51 years or above (*p* < 0.001; LBL, Figure 1).

### 3.3. Variables Correlated with a Higher Sero-Prevalence of HAV in Participants Aged 10 or Less

For the study participants aged 10 or less (*n* = 97), we found that those residing in the Central region had a significantly lower HAV sero-prevalence (4.5%) compared with those residing in the North (40.0%) and the South (66.7%; *p* < 0.001, χ^2^ test). In addition, the participants who used filtered municipal water had a significantly lower HAV sero-prevalence (5.7%) compared with those using unfiltered municipal water (50.0%; *p* = 0.002, χ^2^ test). Within this age group (≤10 years), no significant difference was found between the HAV sero-positive and negative participants (*p* = 0.512; M–W test). Sex and monthly income did not yield statistically significant results when comparing the HAV sero-positive and negative groups (Figure 2).

Neither the number of household inhabitants nor the 3F score showed statistical differences between the HAV sero-positive and negative groups (*p* = 0.876 and 0.478, respectively; M–W).

### 3.4. Variables Correlated with a Higher Sero-Prevalence of HAV in Participants Aged 15 or Less

For the study participants aged 15 or less (*n* = 155), the place of residence was a significant variable in relation with HAV sero-prevalence, with residence in the Central region associated with a lower HAV sero-prevalence (10.0%) compared with the Northern region (25.0%) and Southern region (42.9%; *p* = 0.019, χ^2^ test). Using filtered municipal water sources was associated with a lower sero-prevalence of HAV compared with unfiltered municipal water (*p* = 0.036, χ^2^ test, Table 2). The multinomial logistic regression analysis showed that negativity for HAV IgG was correlated with reporting the use of filtered municipal water compared with unfiltered municipal water (odds ratio: 3.4, 95% CI: 1.2–9.4, *p* = 0.019). Increasing age was the sole covariate associated with a higher sero-positivity for HAV (*p* < 0.001).

A higher number of household inhabitants was found among the HAV sero-positive group compared with the negative group (mean: 6.5 vs. 5.6; *p* = 0.043, M–W). The 3F score did not show a significant difference between the HAV sero-positive and negative groups (*p* = 0.423; M–W).

### 3.5. Variables Correlated with a Higher Sero-Prevalence of HAV in Participants Aged 30 or Less

For the study participants aged 30 or less (*n* = 274), residence in the Central region was associated with a lower HAV sero-prevalence (18.9%) compared with residence in the Northern region (37.5%) and Southern region (44.4%; *p* = 0.042, χ^2^ test). Advancing in age was also associated with a significant increase in the sero-prevalence of HAV from 13.3% among those aged (3–5 years), reaching 85.7% among those aged (21–30 years; *p* < 0.001, χ^2^ test, Table 3). Neither the number of household inhabitants nor the 3F score showed statistical differences between the HAV sero-positive and negative groups (*p* = 0.373 and 0.203, respectively; M–W). Multinomial logistic regression analysis did not display a significant correlation between the negativity for HAV IgG and the source of drinking water (filtered municipal water vs. unfiltered municipal water, *p* = 0.062).

### 3.6. Demonstration of an Epidemiologic Shift in HAV Sero-Prevalence in Jordan between 2008 and 2021

To investigate the occurrence of an epidemiologic shift in HAV sero-prevalence in Jordan, we compared our results to those of 2008 HAV sero-prevalence study in Jordan based on the same age categorization (Figure 3). The comparison displayed a significant decrease in HAV sero-prevalence in each age category (*p* < 0.001; χ^2^ test, Figure 3), with the exception of the age group of 2–4 years, where no statistically significant difference was noticed between the two time periods.

## 4. Discussion

The major result of this study is the demonstration of an epidemiologic shift in HAV sero-prevalence over a decade. In this study, the overall sero-prevalence of HAV was determined at 38.3%. This result points to an intermediate- to low-endemicity of HAV in Jordan, which is in line with the previous study in the country that was conducted in 2008 [26]. The previous comprehensive report by Hayajneh et al. used about 3000 samples that were collected in 2008 and reported an intermediate endemicity of HAV in Jordan, with an overall HAV sero-prevalence of 51% [26]. The evaluation of HAV epidemiology in Jordan can be tracked back to 1980s, with the results pointing to a high endemicity of HAV in the country (100% sero-prevalence by age of 5 years) at that time [28]. In 2004, a report that investigated HAV incidence in the capital Amman was published, which involved more than 1000 individuals with sample collection from January 1991 to December 2001, and this report found an incidence of HAV infection between 1.1–9.6 cases per 100,000 population [25]. The latest report by Hayajneh et al. and our findings suggest the ongoing shift in HAV epidemiology in Jordan towards the low-endemicity category [26]. A plausible explanation for this epidemiologic shift can be related to the continuous improvements in sanitation and hygienic conditions in Jordan.

An important point should be emphasized in relation to the timing of our study. This is related to the introduction of HAV vaccination in Jordan as part of the national vaccination schedule in the country starting from July 2020. Thus, our study can be viewed as the baseline reference point before the implementation of HAV vaccination and can form the basis for future analyses investigating the impact of vaccination in the country on HAV epidemiology. Additionally, the results of this study clearly point to the timely introduction of HAV vaccination in Jordan, as well as its potential cost benefits considering the risks associated with the aforementioned epidemiologic shift towards a low endemicity category [27].

Comparisons of our results with other studies in the Middle East and North Africa (MENA) region are important, as Jordan is part of this region, which has witnessed several migration waves as a result of the ongoing political instability and civil wars witnessed in the last two decades [29]. A recent review by Mehmet Koroglu et al. showed the heterogeneous patterns of HAV endemicity in different MENA countries [30]. In this review, Jordan was classified as an HAV high-endemicity country based on the age at midpoint of population immunity (AMPI) analysis [30]. AMPI can be defined as the youngest age with at least 50% of the population positive for HAV IgG, indicating previous exposure to the virus [18]. The review by Koroglu et al. found that AMPI for HAV in Jordan was 11 years based on the previous serologic survey that was conducted in 2008, compared with an estimated AMPI of between 21 and 30 years in this study [26,30]. A possible explanation for this result, besides the potential epidemiologic shift of HAV epidemiology, is that the majority of samples in this study came from individuals living in the Central region of Jordan (particularly in Amman). A closer look at the study by Hayajneh et al. reveals the higher sero-prevalence of HAV in a majority of other governorates in Jordan at a younger age [26]. Thus, our results indicate the need for prioritizing the Central region for HAV vaccination, considering the lower sero-prevalence of HAV and the accompanying potential for wide-scale outbreaks if no preventive actions are conducted in the region.

Furthermore, the previously mentioned review showed that the MENA countries neighbouring Jordan are classified as either high-endemicity countries (Lebanon) or very high-endemicity countries (Iraq, Palestine, Syria, and Egypt) [31,32,33,34,35]. However, careful interpretation of such classifications should be considered by taking into account the different time points used to conduct HAV sero-surveys in different MENA countries. Thus, the previous recommendation by Yassin et al. that an HAV vaccination program is not needed in Palestine should be updated in light of a more recent and comprehensive HAV sero-prevalence study in the country [33].

The conspicuous epidemiologic shift in HAV sero-prevalence with a decline in HAV IgG positivity in younger age groups was also reported in a review by Melhem et al. in the MENA countries [36]. Thus, the observation of such a decline in this study appears as an expected result and puts a further emphasis on the possible benefit of HAV vaccination in Jordan [27]. This is particularly important in Jordan, considering the current instabilities in the region, which was accompanied by population displacements involving refugees settling in heavily populated camps with poor sanitation and access to clean drinking water [37,38,39]. The recommendation of HAV vaccination in the region was also reported in a recent review involving the Eastern Mediterranean region [40].

The consequences of the declining sero-prevalence of HAV without concomitant implementation of HAV vaccination can involve the occurrence of outbreaks at a large scale. This is particularly seen in the populations of different developed countries in the context of traveling to highly endemic areas. Examples of such outbreaks involved recent history of travel from different European countries to the MENA region [41,42,43]. Such a risk highlights the importance of monitoring the epidemiology of hepatitis A and the benefits of vaccination in regions with declining HAV sero-prevalence rates [44].

Another important finding in this study is the observation that the source of drinking water was correlated with HAV sero-prevalence among individuals of the younger age group (<15 years). Access to clean drinking water has been linked to reduced transmission rates of HAV [45]. However, the previous review in the MENA region by Koroglu et al. found no such correlation [30]. In this study, the classification of municipal water (the major source of drinking water among the study population) into filtered vs. unfiltered was based on the assumption that a lack of water filtration could lead to the use of potentially contaminated water. This can be explained by the fact that Jordan is the second most water scarce country in the world. Consequently, water is supplied once a week, with less availability during summer months [46,47]. Additionally, cross-contamination of the stored water in tanks can result from old and leaking pipes, which should be further investigated in future studies. In this study, the participants were asked about their main source of drinking water, and municipal water represented a major source. However, with a shortage in water supply, the reliance on supplies of potentially untreated water (e.g., springs and wells) increases. Thus, these sources should be monitored by governmental agencies to ensure its safety, as well as the need for spreading messages emphasizing the importance of regular maintenance of water storage places.

A major limitation in this study is related to the sampling approach. We utilized a healthy/patient convenience sampling method that involved a single site (JUH, which is a tertiary care teaching hospital in Amman, serving individuals that reside mainly in the Central region of the country). Thus, sampling bias would be inevitable in this study and should be considered carefully in the interpretation of results. However, the Central region in Jordan harbours more than two-thirds of the country’s population, which can provide a clue that the results of this study might be representative of a large part of Jordan.

Another limitation of this study was the timing of sample collection, which coincided with the COVID-19 pandemic and with its associated restriction of population movement. This was one of the motivations to use a convenience sample; thus, future research investigating the epidemiology of HAV in the country should consider the utility of a refined sampling approach at a national level. The small sample size could be considered as another caveat; however, the current study represents the initial report from an ongoing study that could form the baseline reference point for evaluation of the impact of HAV vaccination introduction in Jordan. Thus, future studies investigating the epidemiology of hepatitis in Jordan should consider a larger sample size with random sampling at a national level to confirm the findings of this research. Furthermore, analysis of the variables that could be linked to a higher sero-prevalence of HAV IgG distribution resulted in a division of the study sample into smaller groups, which should be considered carefully in attempts to evaluate such results and should be addressed in follow-up studies.

## 5. Conclusions

An epidemiologic shift in HAV sero-prevalence was noticed among the study population with a lower positivity for HAV IgG among younger age groups compared with previous studies in the country and the region. This can be related to the continuous improvements in sanitation services and hygienic measures. This downward shift towards a lower endemicity of HAV in Jordan might be accompanied by potential susceptibility to a higher frequency of symptomatic disease and outbreaks; hence, the HAV vaccine inclusion in the national vaccination program is a step in the right direction and its public health benefits should be evaluated.

## Figures and Tables

**Figure 1 pathogens-10-01081-f001:**
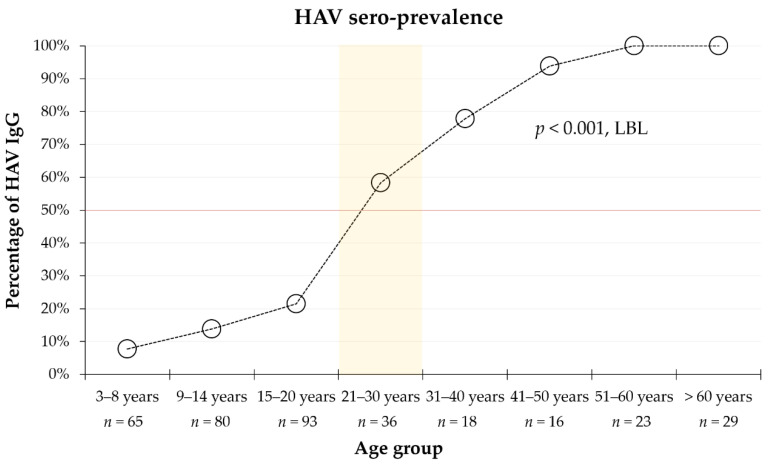
The sero-prevalence of HAV among the study participants divided by different age groups. HAV—hepatitis A virus; IgG—immunoglobulin G; LBL—linear-by-linear test for association. The predicted age at the midpoint of population immunity is highlighted by the light orange rectangle, while the red line represents 50% HAV IgG prevalence; *n*—number of individuals in each age group.

**Figure 2 pathogens-10-01081-f002:**
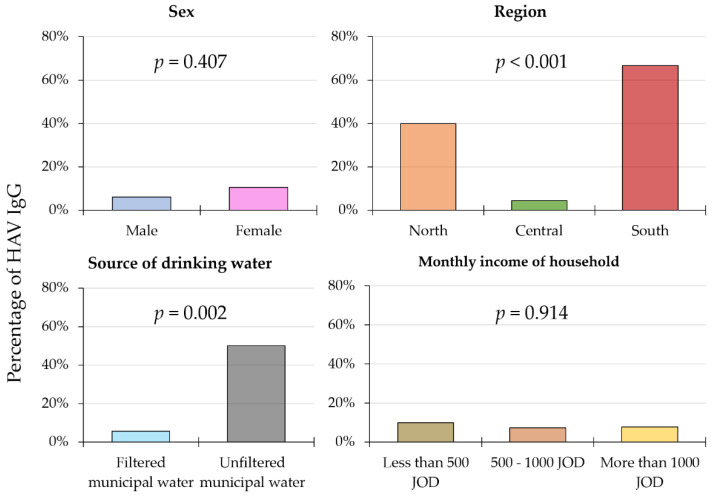
Variables correlated with a higher sero-prevalence of HAV in participants aged 10 or less. HAV—hepatitis A virus; IgG—immunoglobulin G; JOD—Jordanian Dinar; *p*-values are calculated using chi-squared test; for drinking water source, a comparison is done for participants who use municipal water only.

**Figure 3 pathogens-10-01081-f003:**
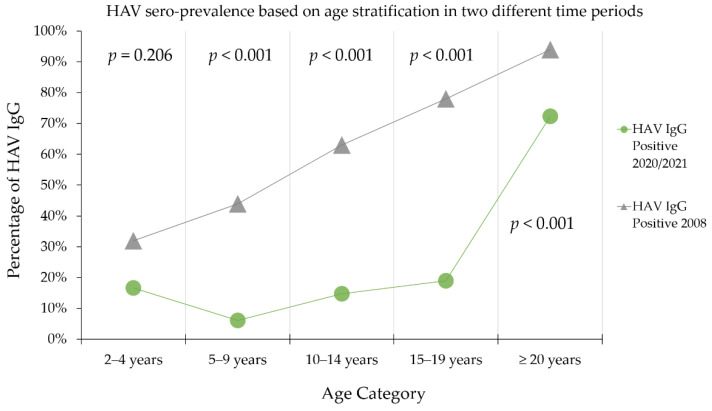
Comparison of HAV IgG sero-prevalence between 2008 and 2020/2021 using the same age categorization. *p*-values to assess differences in HAV IgG sero-prevalence in each age category between the two periods calculated using the chi-squared test. HAV—hepatitis A virus; IgG—immunoglobulin G; The source of 2008 data is available in the study by Hayajneh et al. [26]. The permission was granted by John Wiley & Sons, Inc., License Number: 5135980527369.

**Table 1 pathogens-10-01081-t001:** General features of the study participants.

Variable		Number	Percentage
**Age in years (mean, SD ^1^)**		23.7 (19.6)	
**Number of household inhabitants (mean, SD)**		5.4 (1.8)	
**Sex**	*Male*	148	41.1%
	*Female*	212	58.9%
**Nationality**	*Jordanian*	353	98.1%
	*Non-Jordanian*	7	1.9%
**Region ^2^**	*North*	19	5.3%
	*Central*	332	92.2%
	*South*	9	2.5%
**Monthly income of household**	*Less than 500 JOD ^3^*	118	32.8%
	*500–1000 JOD*	156	43.3%
	*More than 1000 JOD*	86	23.9%
**Educational level**	*High school or less*	294	81.7%
	*Undergraduate study*	54	15.0%
	*Postgraduate study*	12	3.3%
**Drinking water source**	*Mineral water*	5	1.4%
	*Municipal water filtered at home*	124	34.4%
	*Filtered by a local provider*	199	55.3%
	*Unfiltered municipal water*	27	7.5%
	*Stream or well water*	5	1.4%

^1^ SD—standard deviation; ^2^ Region: the Central region includes the following governorates: Balqa, Amman, Zarqa, and Madaba; the Northern region includes the following governorates: Irbid, Ajloun, Jerash, and Mafraq; and the Southern region includes the following governorates: Karak, Tafilah, Ma’an, and Aqaba; ^3^ JOD—Jordanian Dinar.

**Table 2 pathogens-10-01081-t002:** Variables correlated with a higher sero-prevalence of HAV in participants aged 15 or less.

Variable	*Category*	HAV IgG Positive		HAV IgG Negative		*p*-Value ^5^
		N ^4^	%	N	%	
**Sex**	*Male*	7	9.1%	70	90.9%	0.232
	*Female*	12	15.4%	66	84.6%	
**Region ^1^**	*North*	2	25.0%	6	75.0%	0.019
	*Central*	14	10.0%	126	90.0%	
	*South*	3	42.9%	4	57.1%	
**Monthly income**	*Less than 500 JOD ^3^*	5	10.6%	42	89.4%	0.457
	*500–1000 JOD*	11	15.7%	59	84.3%	
	*More than 1000 JOD*	3	7.9%	35	92.1%	
**Drinking water source ^2^**	*Filtered municipal water*	5	9.8%	46	90.2%	0.036
	*Unfiltered municipal water*	4	33.3%	8	66.7%	
**Age**	*1–5 years*	4	13.3%	26	86.7%	0.115
	*6–10 years*	4	6.0%	63	94.0%	
	*11–15 years*	11	19.0%	47	81.0%	

^1^ Region: The Central region includes the following governorates: Balqa, Amman, Zarqa, and Madaba; the Northern region includes the following governorates: Irbid, Ajloun, Jerash, and Mafraq; and the Southern region includes the following governorates: Karak, Tafilah, Ma’an, and Aqaba; ^2^ Drinking water source: The comparison was done for participants who use municipal water only; ^3^ JOD—Jordanian Dinar; ^4^ N—number; ^5^
*p*-values: Calculated using chi-squared test.

**Table 3 pathogens-10-01081-t003:** Variables correlated with a higher sero-prevalence of HAV in participants aged 30 or less.

Variable	*Category*	HAV IgG Positive		HAV IgG Negative		*p*-Value ^5^
		N ^4^	%	N	%	
**Sex**	*Male*	17	15.7%	91	84.3%	0.096
	*Female*	40	24.1%	126	75.9%	
**Region ^1^**	*North*	6	37.5%	10	62.5%	0.042
	*Central*	47	18.9%	202	81.1%	
	*South*	4	44.4%	5	55.6%	
**Monthly income**	*Less than 500 JOD ^3^*	14	17.9%	64	82.1%	0.594
	*500–1000 JOD*	30	23.4%	98	76.6%	
	*More than 1000 JOD*	13	19.1%	55	80.9%	
**Drinking water source ^2^**	*Filtered municipal water*	15	16.3%	77	83.7%	0.093
	*Unfiltered municipal water*	6	33.3%	12	66.7%	
**Age**	*3–5 years*	4	13.3%	26	86.7%	<0.001
	*6–10 years*	4	6.0%	63	94.0%	
	*11–15 years*	11	19.0%	47	81.0%	
	*16–20 years*	17	20.5%	66	79.5%	
	*21–25 years*	9	40.9%	13	59.1%	
	*26–30 years*	12	85.7%	2	14.3%	

^1^ Region: The Central region includes the following governorates: Balqa, Amman, Zarqa, and Madaba; the Northern region includes the following governorates: Irbid, Ajloun, Jerash, and Mafraq; and the Southern region includes the following governorates: Karak, Tafilah, Ma’an, and Aqaba. ^2^ Drinking water source: The comparison was done for participants who use municipal water only; ^3^ JOD—Jordanian Dinar; ^4^ N—number; ^5^
*p*-values—Calculated using chi-squared test.

## Data Availability

The data that support the findings of this study are available from the corresponding author (M.S.) upon reasonable request. The source of 2008 data is available in the study by Hayajneh et al. [26]. The permission was granted by John Wiley & Sons, Inc., License Number: 5135980527369, License date: 25 August 2021.

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
