# Peer review of "Tracking the Epidemiologic Shifts in Hepatitis A Sero-Prevalence Using Age Stratification: A Cross-Sectional Study at Jordan University Hospital"

_pathogens, 2021, doi:10.3390/pathogens10091081_

Round 1

Reviewer 1 Report

The investigators have evaluated the seroepidemiology of hepatitis A virus infection in the general population. I suggest to perform a multivariate analysis in order to make a better evaluation regarding positive serologic status for anti-HAV antibody and independent variables such as age, region, gender, water source and others (in other words, the variables included in table 3 of the manuscript). Another important shortcoming of the study is that the size of the study group is limited (n=360 individuals). 

Author Response

Reviewer #1 comments

The investigators have evaluated the seroepidemiology of hepatitis A virus infection in the general population. I suggest to perform a multivariate analysis in order to make a better evaluation regarding positive serologic status for anti-HAV antibody and independent variables such as age, region, gender, water source and others (in other words, the variables included in table 3 of the manuscript).

Response: We would to thank the reviewer for this valuable comment, however, we believe that we conducted the required analysis, which can be found in the following pages: Page 6, lines 228 – 232; and Page 7, lines 259 – 261.

Another important shortcoming of the study is that the size of the study group is limited (n=360 individuals). 

Response: We would to thank the reviewer for this important comment. However, we pointed to the issue of sample size as a limitation in our study (Page 10, lines 371 – 373). Additionally, we believe that this study can be helpful as an initial report evaluating the status of HAV sero-prevalence (at least in Amman), which can motivate a follow-up nationwide study that addressed the limitations of the current research.

Additionally, calculation of the minimum sample size required to estimate the prevalence indicated a minimum sample size of 282 based on the following calculations that were added to the methods section: Sample size determination was based on calculations done via Epitools - Epidemiological Calculators available freely online (https://epitools.ausvet.com.au/). The minimum required sample size was determined at 282 based on the following parameters in “Sample size to estimate true prevalence”: Assumed true prevalence=50.0%, sensitivity and specificity=0.99, confidence level=0.90, and precision=0.05

Reviewer 2 Report

The study by  Nariman Kareem..... et al. has some epidemiological interest on the seroprevalence of  HAV. However,  the main limitation of this study is the very low number of individuals (N = 360), it is very small for this type of study, in addition to the bias that represents that 65% are residents of the city of Aman. This makes it difficult to consider these results as an epidemiological study, thus concluding that the epidemiology of this infection represents the country as a whole, which is what is intended in the study. In fact, this limitation is already indicated in the discussion on page 9 (line 317). The manuscript cites a previous study with more than 3000 individuals, which would be the cohort size necessary to give solid results. Other additional limitations are:

1-The distribution by age or in relation to water sources results in even smaller groups, so the conclusions of the study are not reliable.

2-The reported seroprevalence does not provide much additional information to that already known in previous studies, such as the one in citation 26

3-The results of seroprevalence by regions (3.4 ;, figure 2, Table 2) are highly questionable considering the great differences between the corresponding populations: North 19; Central 332 and Sur 9. This same limitation is found in the results related to the water source, which, although they are probably correct, since they were expected, do not allow us to confirm the conclusions by themselves.

4-The data included in section 3.6 (figure 3) are interesting although they have the general limitation of the study, which is the small population studied. In addition, the origin of the seroprevalence data in 2008 must be referenced (citation 26?),

Author Response

Reviewer #2 comments

The study by Nariman Kareem..... et al. has some epidemiological interest on the seroprevalence of HAV. However, the main limitation of this study is the very low number of individuals (N = 360), it is very small for this type of study, in addition to the bias that represents that 65% are residents of the city of Aman. This makes it difficult to consider these results as an epidemiological study, thus concluding that the epidemiology of this infection represents the country as a whole, which is what is intended in the study. In fact, this limitation is already indicated in the discussion on page 9 (line 317). The manuscript cites a previous study with more than 3000 individuals, which would be the cohort size necessary to give solid results.

Response: We would like to thank the reviewer for the valuable comment regarding potential bias in the study sample in relation to a majority of samples from the Central region in Jordan and in relation to the small sample size. However, we indicated in the study title and the objective of the study (Page 2, lines 97 – 101) that the current study focused mainly on the visitors of Jordan University Hospital in Amman, which serves individuals residing mainly in the Central region of the country. The value of this study can be seen in relation to the low number of similar studies in the region, and it can form the basis for more comprehensive work and motivate more studies in the region to investigate HAV epidemiology while addressing its limitations.

Additionally, calculation of the minimum sample size required to estimate the prevalence indicated a minimum sample size of 282 based on the following calculations that were added to the methods section: Sample size determination was based on calculations done via Epitools - Epidemiological Calculators available freely online (https://epitools.ausvet.com.au/). The minimum required sample size was determined at 282 based on the following parameters in “Sample size to estimate true prevalence”: Assumed true prevalence=50.0%, sensitivity and specificity=0.99, confidence level=0.90, and precision=0.05

Other additional limitations are:

1-The distribution by age or in relation to water sources results in even smaller groups, so the conclusions of the study are not reliable.

Response: We would like to thank the reviewer for the comment; however, we elaborated on these potential limitations in the discussion section (Pages 9 and 10, lines 360 – 373). Nevertheless, and based on the reviewer’s important comment, we added the following statement to the limitations paragraphs: “Furthermore, analysis of the variables that could be linked to a higher sero-prevalence of HAV IgG distribution resulted in division of the study sample into smaller groups, which should be considered carefully in the attempts to evaluate such results and should be addressed in the follow-up studies.” Page 10, line 373 – 376.

2-The reported seroprevalence does not provide much additional information to that already known in previous studies, such as the one in citation 26

Response: We would like to thank the reviewer for this comment; however, we disagree on this point. As you can check in Figure 3, comparison using the same age stratification with the 2008 study by Hayajneh et al, shows that for every single age group older than four years, a significant decline in the sero-prevalence of HAV IgG was noticed compared to our results which clearly displays the significant decline demonstrated in this study.

3-The results of seroprevalence by regions (3.4 ;, figure 2, Table 2) are highly questionable considering the great differences between the corresponding populations: North 19; Central 332 and Sur 9. This same limitation is found in the results related to the water source, which, although they are probably correct, since they were expected, do not allow us to confirm the conclusions by themselves.

Response: We would like to thank the reviewer for the comment; however, we have already discussed the limitations of this study in relation to potential sample bias. Additionally, the fact that two-thirds of the Jordanian population reside in the Central region of the country can point to potential benefit of the study results at least in this region of the country.

4-The data included in section 3.6 (figure 3) are interesting although they have the general limitation of the study, which is the small population studied. In addition, the origin of the seroprevalence data in 2008 must be referenced (citation 26?)

Response: We would like to thank the reviewer for spotting this missing reference and based on the reviewer’s comment, we added the following statement to the figure footnote as follows: “The source of 2008 data is available in the study by Hayajneh et al [26].”

Additionally, this statement was also added to the data availability statement in Page 10, lines 406 – 407.

Reviewer 3 Report

Kareem et al describe the seroprevalence of hepatitis A virus in children and young adults in Jordan. The study is well conducted and the results are clearly described. The authors provide interesting data on the current seroprevalence of HAV that might be useful to Health Authorities in order to decide on vaccination strategies. The manuscript may benefit from some minor clarifications.

  • Introduction, line 67: The authors may like to clarify that the detection of anti-HAV IgG is useful to evaluate endemicity in “non-vaccinated”
  • Material and Methods line 106: There is no sample size calculation. Are the individual included in the study all individuals that attended the hospital and agreed to participate in the study? Is there any bias in the inclusion criteria? What do the author mean by “convenience sampling”?
  • Material and Methods line 127: The authors refer to a questionnaire attached as Appendix A, which is not available for review. Does this questionnaire include a question on HAV vaccine? Are only non-vaccinated individuals included in the study?
  • Figure 1: The sample size for each age group should be included.
  • Discussion: Authors may like to comment on the usefulness of a combined vaccine for HAV and HBV in Jordan. Is there a vaccination program for HBV implemented in the country? Is HBV a relevant health problem in Jordan? If so, would HAV vaccination program facilitate HBV vaccination in the future?

Author Response

Reviewer #3 comments

Kareem et al describe the seroprevalence of hepatitis A virus in children and young adults in Jordan. The study is well conducted and the results are clearly described. The authors provide interesting data on the current seroprevalence of HAV that might be useful to Health Authorities in order to decide on vaccination strategies. The manuscript may benefit from some minor clarifications.

We would like to thank the reviewer for the positive comments

Introduction, line 67: The authors may like to clarify that the detection of anti-HAV IgG is useful to evaluate endemicity in “non-vaccinated”

Response: We would like to thank the reviewer for this valuable comment and accordingly, we added the following statement to the introduction section, Page 2, lines 68 – 69: “The World Health Organization (WHO) advocates two approaches to measure HAV sero-prevalence, which are based on the detection of anti-HAV immunoglobulin G (IgG) anti-bodies to determine the endemicity of HAV in countries with a majority of non-vaccinated individuals [19].”

Material and Methods line 106: There is no sample size calculation. Are the individual included in the study all individuals that attended the hospital and agreed to participate in the study? Is there any bias in the inclusion criteria? What do the author mean by “convenience sampling”?

Response: We would like to thank the reviewer for this valuable comment and accordingly, we added the following statement to the methods section, Page 4, lines 171 – 175: “Sample size determination was based on calculations done via Epitools - Epidemiological Calculators available freely online (https://epitools.ausvet.com.au/). The minimum required sample size was determined at 282 based on the following parameters in “Sample size to estimate true prevalence”: Assumed true prevalence=50.0%, sensitivity and specificity=0.99, confidence level=0.90, and precision=0.05”

Moreover, in the methods section the recruitment process was described and was based on the individuals who visited JUH and agreed to participate in the study. Bias was inevitable particularly in relation to site of residence and this was mentioned in the limitations section. Convenience sampling referred to the sampling approach where we included the readily available participants.

Material and Methods line 127: The authors refer to a questionnaire attached as Appendix A, which is not available for review. Does this questionnaire include a question on HAV vaccine? Are only non-vaccinated individuals included in the study?

Response: Regarding the missing Appendix, we will make sure it will be uploaded in the revision files, and regarding the vaccination for HAV, no such question was included and the reason for its absence is the lack of HAV vaccination in Jordan prior to July 2020, so we are inclined to believe that the vast majority of participants in this study did not receive HAV vaccination.

Figure 1: The sample size for each age group should be included.

Response: Based on the reviewer’s comment, we edited the figure by adding the number of individuals in each age group.

Discussion: Authors may like to comment on the usefulness of a combined vaccine for HAV and HBV in Jordan. Is there a vaccination program for HBV implemented in the country? Is HBV a relevant health problem in Jordan? If so, would HAV vaccination program facilitate HBV vaccination in the future?

Response: We are inclined to leave the focus on HAV since HBV vaccination in Jordan has been available from late 1990s and to keep the focus on HAV.

Round 2

Reviewer 2 Report

After the responses provided by authors to my constraitns in relation to this manuscript . I want to comment that the answers and modification related to my comments 1 to 4 are now ok for me . However, I have still some constraints in relation to my main comment about the size of the population, also commented by other reviewer, to which the authors say:

"We would like to thank the reviewer for the valuable comment regarding potential bias in the study sample in relation to a majority of samples from the Central region in Jordan and in relation to the small sample size. However, we indicated in the study title and the objective of the study (Page 2, lines 97 – 101) that the current study focused mainly on the visitors of Jordan University Hospital in Amman, which serves individuals residing mainly in the Central region of the country. The value of this study can be seen in relation to the low number of similar studies in the region, and it can form the basis for more comprehensive work and motivate more studies in the region to investigate HAV epidemiology while addressing its limitations.

Additionally, calculation of the minimum sample size required to estimate the prevalence indicated a minimum sample size of 282 based on the following calculations that were added to the methods section: Sample size determination was based on calculations done via Epitools - Epidemiological Calculators available freely online (https://epitools.ausvet.com.au/). The minimum required sample size was determined at 282 based on the following parameters in “Sample size to estimate true prevalence”: Assumed true prevalence=50.0%, sensitivity and specificity=0.99, confidence level=0.90, and precision=0.05"

I do not consider enough this explanation because as the authors have included in their own results the average prevalence is less than 50%  (average 38%). Therefore, a higher size of the population than indicated would be needed for validate their results.  However, I agree with the authors than despite these data, this study could have certain value in relation to the low number of similar studies in the region. 

Author Response

We would to thank the reviewer for the previous comments that enabled the improvement of our manuscript.

Again, we are thankful for the reviewer for this valuable comment, and based on the comment we added the following statement to the limitations (Page: 11, lines 282-284): “Thus, the future studies investigating the epidemiology of hepatitis in Jordan should consider a larger sample size with random sampling at the national level to confirm the findings of this research.”